# Nebulized Heparin in Burn Patients with Inhalation Trauma—Safety and Feasibility

**DOI:** 10.3390/jcm9040894

**Published:** 2020-03-25

**Authors:** Gerie J. Glas, Janneke Horn, Jan M. Binnekade, Markus W. Hollmann, Jan Muller, Berry Cleffken, Kirsten Colpaert, Barry Dixon, Nicole P. Juffermans, Paul Knape, Marcel M. Levi, Bert G. Loef, David P. Mackie, Manu L.N.G. Malbrain, Benedikt Preckel, Auke C. Reidinga, K.F. van der Sluijs, Marcus J. Schultz

**Affiliations:** 1Laboratory of Experimental Intensive Care and Anesthesiology (L·E·I·C·A), Amsterdam University Medical Centers, 1105 AZ Amsterdam, The Netherlands; j.horn@amc.uva.nl (J.H.); m.w.hollmann@amc.uva.nl (M.W.H.); n.p.juffermans@amc.uva.nl (N.P.J.); b.preckel@amc.uva.nl (B.P.); marcus.j.schultz@gmail.com (M.J.S.); 2Department of Anesthesiology, Amsterdam University Medical Centers, 1105 AZ Amsterdam, The Netherlands; 3Department of Intensive Care, Amsterdam University Medical Centers, 1105 AZ Amsterdam, The Netherlands; j.m.binnekade@amc.uva.nl; 4Department of Intensive Care, University Hospital Gasthuisberg, 3000 Leuven, Belgium; jan.muller@uzleuven.be; 5Department of Intensive Care, Maasstad Hospital, 3079 DZ Rotterdam, The Netherlands; cleffkenb@maasstadziekenhuis.nl; 6Department of Intensive Care, Ghent University Hospital, 9000 Ghent, Belgium; kirsten.colpaert@ugent.be; 7Department of Intensive Care, St Vincent’s Hospital, Melbourne 3065, Australia; Barry.DIXON@svha.org.au; 8Department of Intensive Care, Red Cross Hospital, 1942 LE Beverwijk, The Netherlands; pknape@rkz.nl (P.K.); sd.mackie@wxs.nl (D.P.M.); 9Department of Vascular Medicine, Amsterdam University Medical Centers, 1105 AZ Amsterdam, The Netherlands; m.m.levi@amc.uva.nl; 10Department of Intensive Care, Martini Hospital, 9728 NT Groningen, The Netherlands; B.Loef@mzh.nl (B.G.L.); A.C.Reidinga@mzh.nl (A.C.R.); 11Department of Intensive Care and Faculty of Medicine and Pharmacy, University Hospital Brussels, Jette, Belgium and Free University of Brussels, 1090 Brussels, Belgium; manu.malbrain@uzbrussel.be

**Keywords:** burn, inhalation trauma, heparin, nebulization, safety

## Abstract

Background: Pulmonary hypercoagulopathy is intrinsic to inhalation trauma. Nebulized heparin could theoretically be beneficial in patients with inhalation injury, but current data are conflicting. We aimed to investigate the safety, feasibility, and effectiveness of nebulized heparin. Methods: International multicenter, double-blind, placebo-controlled randomized clinical trial in specialized burn care centers. Adult patients with inhalation trauma received nebulizations of unfractionated heparin (25,000 international unit (IU), 5 mL) or placebo (0.9% NaCl, 5 mL) every four hours for 14 days or until extubation. The primary outcome was the number of ventilator-free days at day 28 post-admission. Here, we report on the secondary outcomes related to safety and feasibility. Results: The study was prematurely stopped after inclusion of 13 patients (heparin *N* = 7, placebo *N* = 6) due to low recruitment and high costs associated with the trial medication. Therefore, no analyses on effectiveness were performed. In the heparin group, serious respiratory problems occurred due to saturation of the expiratory filter following nebulizations. In total, 129 out of 427 scheduled nebulizations were withheld in the heparin group (in 3 patients) and 45 out of 299 scheduled nebulizations were withheld in the placebo group (in 2 patients). Blood-stained sputum or expected increased bleeding risks were the most frequent reasons to withhold nebulizations. Conclusion: In this prematurely stopped trial, we encountered important safety and feasibility issues related to frequent heparin nebulizations in burn patients with inhalation trauma. This should be taken into account when heparin nebulizations are considered in these patients.

## 1. Introduction

Inhalation trauma is a major cause of morbidity and mortality in burn patients [1,2,3]. It is associated with respiratory failure, pulmonary infections, and prolonged mechanical ventilation [1].

Currently, there is no therapeutic intervention available for inhalation trauma and management consists of supportive care [3,4,5]. Mechanical ventilation can be challenging due to the presence of characteristic widespread fibrin casts which obstruct the airways and subsequently promote regional barotrauma and ventilation/perfusion mismatch [3,6]. This concept led to preclinical and clinical studies investigating the use of nebulized anticoagulants, mainly heparin [7]. Nebulized heparin, alone or combined with other agents, attenuated lung injury and improved outcome in several preclinical models of inhalation trauma [5]. However, clinical studies investigating the efficacy of nebulized heparin showed conflicting results [7]. We hypothesized nebulized heparin to be safe and to improve clinical outcome in burn patients suffering from inhalation trauma. To investigate this, we initiated a multicenter double-blind placebo-controlled trial.

## 2. Materials and Methods

This international multicenter, double-blind, placebo-controlled randomized clinical trial was performed in specialized burn care centers in the Netherlands and Belgium. The study was registered with trial identification number NCT01773083, and the protocol was published [8] and approved by the Institutional Review Boards of participating centers.

### 2.1. Patients

Adult mechanically ventilated patients with inhalational trauma confirmed by bronchoscopy [9] were eligible for inclusion. Patients were excluded if they could not be enrolled within 36 h after trauma, were mechanically ventilated more than 24 h prior to trial enrollment, had burn injury for ≥ 60% of the total body surface area (TBSA), or were not expected to survive for more than 72 h. Other exclusion criteria included expected duration of mechanical ventilation of less than 24 h, pregnancy or breast feeding, history of severe chronic obstructive pulmonary disease (e.g., with noninvasive ventilation or oxygen therapy at home, or need for frequent systemic corticosteroid therapy for exacerbations), pulmonary bleeding in the past 3 months, history of a significant bleeding disorder, or a known allergy to heparin. Written informed consent from the patient or legal representative was obtained before randomization. Further details are available in our previously published study protocol [8].

### 2.2. Study Medication and Procedures

ALEA^®^ software (TenALEA consortium, Amsterdam, the Netherlands) was used to randomly assign patients to nebulization of identically packed unfractionated sodium heparin (25,000 IU in 5 mL, Pfizer, Melbourne, Australia) or sodium chloride 0.9% (5 mL, Pfizer, Melbourne, Australia), every four hours.

Study medication was nebulized using an AeronebPro system, which was placed on the inspiratory limb of the ventilator circuit. An active humidification system was used, with a filter on the expiratory limb to prevent aerosolized medication from damaging the ventilator’s expiratory valve. The filter was replaced every 24 h. Patients received nebulizations for up to 14 days, successful extubation, or death, whichever came first.

Routine use of mucolytics was not allowed, and attending physicians were advised to use mucolytics only when viscous mucus was considered problematic.

### 2.3. Outcomes

The main outcome was the number of ventilator-free days. This was defined as days of unassisted breathing during the first 28 days after enrollment. Safety, feasibility, and clinical outcomes were secondary outcomes. Safety was assessed using the following endpoints: (1) bleeding attributable to the study drug and occurring during the period of nebulization. This was defined as a bleeding requiring surgery, transfusion of blood products, or any clinically significant bleeding from the lung (requiring bronchoscopy); (2) confirmed heparin-induced thrombocytopenia (HIT); (3) activated partial thromboplastin time (APTT) > 150 s; (4) any transfusion of red blood cells, platelets, or plasma; and (5) other serious adverse events.

Feasibility was assessed based on the study protocol compliance. Data on the number and reason for withheld scheduled dosages were collected. Study medication could be temporarily discontinued for the following reasons: suspected HIT, surgical procedures, clinically relevant bleedings, APTT > 150 s, platelet count ≤ 10 × 10^9^/L, excessive blood in lavage fluids, or sputum. Restart was left at the discretion of the attending physician; however, they were requested to restart as soon as possible [8].

### 2.4. Data Collection

Data collected included: patient characteristics, data on injury severity (including clinical and bronchoscopic inhalational injury severity scores [8]), safety data, and clinical outcomes (including duration of mechanical ventilation and occurrence of pneumonia or acute respiratory distress syndrome (ARDS)) [8]. Length of stay and mortality (intensive care unit (ICU) and hospital) were assessed on days 28 and 90 [8].

### 2.5. Sample Size Calculation

Sample size calculation was based on a trial in which nebulized heparin was associated with a reduction of 4.6 days of mechanical ventilation in critically ill patients expected to require at least 48 h of ventilation [10]. We conservatively estimated a lower improvement in burn patients with inhalation trauma compared to nonburn critically ill patients. A reduction of three days of invasive ventilation was considered as clinically significant. To observe an improvement of three ventilator-free days at day 28 with *p* < 0.05 at 80%, power we required 58 patients per treatment arm [8].

### 2.6. Statistical Analysis

Data were analyzed by the intention-to-treat approach. Continuous variables are presented as median and interquartile range (median (IQR)). Binary and categorical variables are presented as frequencies and percentages (*n* (%)). Comparison between groups was done using Mann-Whitney *U* test. Data were analyzed using SPSS, version 23.0 (SPSS Inc., Chicago, IL, USA).

## 3. Results

In 11 months, we included 13 patients (January 2014–May 2014 and December 2014–July 2015). The study was terminated early on advice of the data safety monitoring board due to low recruitment of patients and high costs associated with the purchase and blinding of trial medication. Therefore, no analyses on efficacy were performed.

In total, 49 patients with suspected inhalation trauma were screened, of which 33 had bronchoscopically confirmed inhalation trauma; thirteen were included (Figure 1).

There were neither statistically significant differences in demographic and baseline characteristics nor outcomes between the groups (Table 1).

### 3.1. Safety

No severe bleedings attributable to the study drug and no cases of HIT were reported. One patient in each group required blood transfusion after severe hemorrhage, which occurred 7 and 13 days after the last nebulization and were not related to a study procedure. The total number of transfused blood products was similar between groups. In the heparin group, 31 units of red blood cells (RBC) were transfused in three patients (1, 5, and 25 units per patient). In the placebo group, 32 units of RBC were transfused in two patients (17 and 15 units per patient). Fresh frozen plasma was transfused in two patients in the heparin group (both received 4 units) compared to one patient in the placebo group. One patient in the heparin group received one unit of platelets.

APTT > 150 s was seen in one patient in the first two days of heparin nebulization, which normalized after discontinuation of nebulizations. During and after the nebulization period, APTT values were similar in patients nebulized with heparin compared to placebo, respectively 36 (IQR 31–43) versus 31 s (IQR 28–37) during a nebulization period and 33 (IQR 29–38) versus 40 s (31–44) after a nebulization period.

### 3.2. Study-Related Serious Adverse Events

Two subsequent reports of serious ventilatory problems prompted a temporary discontinuation of the study. Both reports were related to the nebulization procedure. In one heparin-treated patient, high airway pressures were seen after nebulization with cefazolin. The ventilator was changed to solve the problem. The next nebulization of cefazolin resulted in similar ventilatory problems, which were promptly solved by changing the filter on the expiratory limb of the circuit and cleaning of the expiratory valve. Cefazolin nebulizations were often administered at this site and had not resulted in similar problems before. After each nebulization, the nebulizer was adequately cleaned.

In another patient, there was a rise in peak pressures and no expired minute volume could be measured following the second nebulization with heparin. Tube or airway obstruction by mucus plugging and bronchospasm as well as pneumothorax were ruled out. The high peak pressures were probably caused by an occluded (saturated) filter. The problem persisted after a change of ventilator but was resolved after the ventilator circuit and expiratory filter were changed. Hemodynamic instability developed requiring vasopressors for 20 min. There were no reports of ventilation problems after nebulization procedures the placebo group.

After the second report, the study was temporarily discontinued and restarted after approval of a protocol amendment, standardizing the use of a specific filter on the expiratory limb of the circuit during the period in which the patient was subjected to nebulization of study medication [10]. After this standardization, there were neither new respiratory nor ventilatory adverse events reported.

### 3.3. Other Serious Adverse Events Not Related to the Study

Two in-hospital cardiac arrests were reported, both in the placebo group. The former occurred on the last day of nebulization of study medication in a patient suffering from ARDS and pneumonia. There was a return of spontaneous circulation after six minutes of cardiopulmonary resuscitation (CPR) and the patient recovered. The second cardiac arrest occurred after ICU discharge in a patient ready for discharge to a nursing home. The patient deceased following an unsuccessful CPR attempt. Furthermore, cerebral edema was reported in one patient with traumatic head injury. The remaining serious adverse events (SAEs) took place after the period of nebulizations of the study medication.

### 3.4. Feasibility

In the heparin group, 129 out of 427 scheduled dosages were withheld. One patient considered to be at increased risk of bleeding due to a head trauma accounted for 72 (17%) of dosages withheld. In addition, one patient received two out of 72 scheduled heparin nebulizations as the study was temporarily discontinued for safety reasons following an SAE report.

In the placebo group, 45 out of 299 scheduled dosages were withheld. The main reasons to withhold nebulizations at physicians’ request were blood-stained sputum or bronchoalveolar lavage (BAL) fluid or expected increased bleeding risks (Table 2). Other reasons to withhold study medication included prolonged APTT, surgical procedures, and logistics (Table 2).

## 4. Discussion

In this prematurely stopped randomized controlled trial, there were important safety and feasibility issues related to frequent nebulizations in burn patients with inhalation trauma. Those issues were not the reason for the termination of the study; however, they should be taken into account when heparin nebulizations are considered as therapeutic option and in planning future trials. Serious ventilatory problems during or after nebulization procedures were reported. The combined use of an active humidification system with a filter on the expiratory limb of the ventilator circuit may result in life-threatening obstruction of the filter [11,12]. The expiratory resistance can increase due to filter saturation with water condensates [12]. This risk can be reduced by the use of specific expiratory filters [10]. Other possible complications related to a nebulization procedure include the occurrence of mucus plugging in the artificial airways and clogging of the expiratory limb of the ventilator circuit due to drug precipitation [11,12,13,14].

Inhaled antibiotics have been used for decades to treat lower airway infections after inhalation trauma [15]. Both heparin and placebo nebulization have been tested in combination with other nebulized drugs [4,10,16,17,18]. Therefore, the study nebulizer could be used for other medication, provided that the used drugs were approved for intermittent nebulizations.

There is limited data on the feasibility of heparin nebulization in patients with inhalation trauma, and none of the published retrospective studies reported clearly about treatment adherence [4,17,18,19,20,21]. The only prospective study available reported that heparin nebulization was not stopped in any of the patients [16]. Details about withheld dosages were not presented.

A high number of nebulizations were withheld in our study. Some clinicians were cautious and, at times, reluctant to administer study medication, especially in cases of blood-stained sputum, in planned surgery, or in patients with an expected increased bleeding risk. The clinical significance of blood-stained secretions is a matter of discussion, as blood-stained secretions have been reported in up to 67% of the patients treated with nebulized heparin [10,18], none of which were considered clinically significant. Furthermore, nebulized heparin did not result in significant blood-stained secretions in the only prospective study [16]. Also, two retrospective studies with historical controls reported no increase in clinically significant bleedings for patients nebulized with heparin [18,19].

Dosing and duration of heparin nebulization in our study was based on data from a trial in nonburn critically ill patients in which heparin nebulization was found to be safe and associated with an increase in the number of ventilator-free days [8,10]. Indeed, heparin dosages ranging from 30,000 to 400,000 IU have been used safely in ventilated patients [7]. Notably, low dosages (i.e., 30,000 IU) failed to improve outcomes in most studies in adult burn patients [7]. Although our study used the highest dosage of heparin nebulization for burn patients described in literature thus far, 150,000 IU/day, APTT was only prolonged in one patient. All other APTT values were similar between intervention groups throughout the study period. Moreover, reported APTT are comparable with APTT provided by studies of nebulized heparin in burn patients [16,18].

This study has important limitations. First, the study was terminated early due to the slow inclusion rate and high costs associated with purchase and blinding of study medication. Second, the need for bronchoscopically confirmed inhalation trauma, considered as gold standard for diagnosing inhalation trauma [22], may have contributed to the low inclusion rate. Another major limitation is the large number of withheld nebulizations in spite of frequent communication and regular training emphasizing the importance of a strict protocol adherence. This stresses the impact of institutional bias regarding the safety and possible systemic effects of nebulized heparin.

## 5. Conclusions

This prematurely stopped study addresses important issues of frequent heparin nebulization in adult patients with inhalational trauma. Physicians should be aware of potential complications of the combined use of a nebulizer with active humidification and expiratory limb filters. Moreover, feasibility of frequent heparin nebulizations could be questioned in burn patients considered to have an increased risk of bleeding, requiring simultaneous nebulizations of other agents or requiring prolonged surgical procedures.

## Figures and Tables

**Figure 1 jcm-09-00894-f001:**
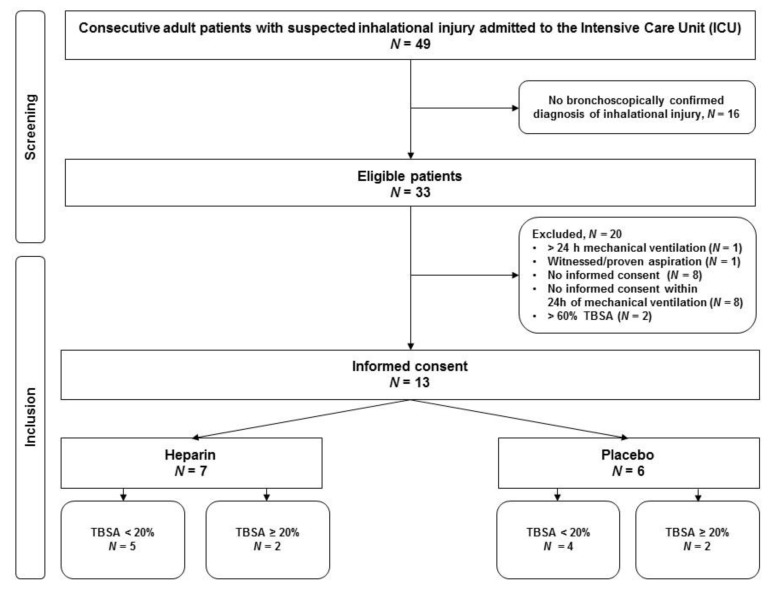
Consort diagram. Abbreviations. TBSA: total body surface area. *N*: number.

**Table 1 jcm-09-00894-t001:** Demographic and baseline characteristics and clinical outcomes.

	Heparin *N* = 7	Placebo *N* = 6	*p*
Age	67 (59–81)	51 (40–61)	0.08
Gender, male	4	4	0.73
TBSA %	12 (1–23)	3 (1–44)	0.88
ABSI	7 (6–9)	6 (5–10)	1.0
Inhalation injury score			0.62
Clinical ^a^	4 (2–4)	3 (3–5)	
Bronchoscopic ^b^	1 (1–2)	1 (1–3)	0.83
SAPS II	51 (37–65)	56 (42–72)	0.83
LIS ^c^ on ICU admission day	1.5 (1–2.3)	1.5 (1–1.8)	0.84
APTT on ICU admission day	34 (27–43)	31 (27–38)	0.51
Ventilator-free days and alive at day 28	16 (4–24)	20 (7–24)	0.62
ICU-free days and alive at day 90	71 (0–75)	49 (3–82)	0.73
ARDS, *N*	-	1	-
Pneumonia, *N*	2	1	-
Burn wound infection, *N*	-	1	-

Data as median (95% interquartile range): There were no statistically significant differences between groups. Abbreviations. ABSI: abbreviated burn severity index. APTT: activated partial thromboplastin time. ARDS: acute respiratory distress syndrome. LIS: lung injury score. *N*: number. SAPS II: Simplified Acute Physiology Score. TBSA: total body surface area. ICU: Intensive Care Unit. ^a^ Clinical scoring of inhalation injury: A score consisting of 7 clinical factors was considered to be suggestive for inhalation injury. One-point-each diagnosis of smoke inhalation is fulfilled with a score > 2. In brief, 1. trapped during a fire in an enclosed space, 2. carbonaceous sputum, 3. altered level of consciousness, 3. mild respiratory distress, 4. serious respiratory distress, and 7. hoarseness or loss of voice. ^b^ Grading of inhalation injury by bronchoscopic criteria (0 = no injury; 1 = mild injury; 2 = moderate injury; 3 = severe injury; and 4 = massive injury). ^c^ Scoring based on chest X-ray findings, PaO_2_/FiO_2_, Positive End Expiratory Pressure (PEEP) level, and respiratory compliance. A score < 2.5 indicates mild-to-moderate lung injury, and ≥ 2.5 indicates severe lung injury.

**Table 2 jcm-09-00894-t002:** Feasibility and reasons to withheld nebulizations.

	Heparin *N* = 7	Placebo *N* = 6
Patients with withheld dosages, *N*	3	2
Number of scheduled/withhold dosages (% withheld)	427/129 (30)	299/45 (15)
Reasons for temporary withholding of scheduled dosages, number of dosages (% of scheduled dosages)		
prolonged APTT (> 150 s)	10 (2)	0
physicians request:		
blood stained sputum or BAL	36 (8)	29 (10)
increased bleeding risk ^1^	72 (17)	0
logistical reasons	4 (1)	0
surgical procedure	3 (1)	10 (3)
reason not specified (protocol violation)	4 (1)	6 (2)
Number of stopped nebulizations ^2^	70	0

Abbreviations. APTT: activated partial thromboplastin time. BAL: broncho alveolar lavage. *N*: number ^1^ One patient with concomitant trauma. ^2^ Nebulizations definitively stopped for safety reasons after a serious adverse event (SAE) report (*N* = 1).

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
