# Peer review of "Nebulized Heparin in Burn Patients with Inhalation Trauma—Safety and Feasibility"

_jcm, 2020, doi:10.3390/jcm9040894_

Round 1
Reviewer 1 Report
Comment 1, Line 34:
…the safety…
Comment 2, Line 64:
Check spelling "…placebo-controlled…"
Comment 3, Line 75:
Check spelling "…randomization…" Keep British English or American English consistently!
Comment 4, Line 84:
Check spelling "…the ventilator´s expiratory valve…"
Comment 5, Line 89:
…during the first 28 days… Please add “after enrollment”
Comment 6, Line 93:
Check spelling "…heparin-induced…"
Comment 7, Line 108:
Statistical analyses: Please add some information about sample size calculation.
Comment 8, Line 112:
…(SPSS Inc., Chicago, IL,)…. (Please add “USA”)
Comment 9, Line 114:
Out of how many planned patients? This can be answered as information about sample size calculation (see comment 7) or within this section.
Comment 10, Line 116:
I wonder why the high costs regarding study medication was one reason for stopping the trial, because I assume that the costs were known at the beginning of the trial. Please specify/clarify.
Comment 11, Line 123:
The capture and legend of figure 1 are missing. Please add.
Comment 12, Table 1:
Please add p values to the table. You declare that there have not been statistical differences, but no p values are reported. In the legend of table 1, “NS.” is mentioned, but “NS.” does not appear in the table.
Comment 13, Line 145:
Check spelling: “temporary discontinuation”
Comment 14, Line 161:
"…the use of a…"
Comment 15, Line 176:
Check spelling (consider “a SAE”)
Authors responses
Comments 1-6 and comments 8, 13-15 consisted of spelling errors:
Comment 1, Line 34: …the safety…
Comment 2, Line 64: Check spelling "…placebo-controlled…"
Comment 3, Line 75: Check spelling "…randomization…" Keep British English or American English consistently!
Comment 4, Line 84: Check spelling "…the ventilator´s expiratory valve…"
Comment 5, Line 89: …during the first 28 days… Please add “after enrollment”
Comment 6, Line 93: Check spelling "…heparin-induced…"
Comment 8, Line 112: …(SPSS Inc., Chicago, IL,)…. (Please add “USA”)
Comment 13, Line 145: Check spelling: “temporary discontinuation”
Answer: Temporarily discontinuation was replaced by ‘temporary discontinuation’.
Comment 14, Line 161: "…the use of a…"
Comment 15, Line 176: Check spelling (consider “a SAE”)
Answer: We rechecked spelling and punctuation throughout the manuscript and corrected the text accordingly.
Comment 7, Line 108: Statistical analyses: Please add some information about sample size calculation.
Answer: we added a paragraph with additional information on our sample size calculation.
‘Sample size calculation
Sample size calculation was based on a trial in which nebulized heparin was associated with a reduction of 4.6 days of mechanical ventilation in critically ill patients expected to require at least 48 hours of ventilation (10). We conservatively estimated a lower improvement in burn patients with inhalation trauma compared to nonburn critically ill patients. A reduction of three days of invasive ventilation was considered as clinically significant. To observe an improvement of three ventilator-free days at day 28 with P < 0.05 at 80% power we required 58 patients per treatment arm (8).’ (Page 3 line 116-122)
Comment 9, Line 114: Out of how many planned patients? This can be answered as information about sample size calculation (see comment 7) or within this section.
Answer: we added a paragraph with information regarding our sample size calculation (see comment 7)
‘To observe an improvement of three ventilator-free days at day 28 with P < 0.05 at 80% power we required 58 patients per treatment arm (8).’ (Page 3 line 121- 122)
Comment 10, Line 116: I wonder why the high costs regarding study medication was one reason for stopping the trial, because I assume that the costs were known at the beginning of the trial. Please specify/clarify.
Answer: The unexpected high costs associated with study medication were related to the low recruitment rate in combination with a short shelf life of heparin, which resulted in wasting of a large amount of expensively labelled study medication.
We did not include this explanation in the manuscript; however, if the reviewer feels that this information is necessary for the reader of the manuscript, we are happy to add this information to the text.
Comment 11, Line 123:
The capture and legend of figure 1 are missing. Please add.
Answer: we added the capture and legend of figure 1.
Figure 1. Consort diagram
Abbreviations. TBSA: total body surface area. N: number
Comment 12, Table 1:
Please add p values to the table. You declare that there have not been statistical differences, but no p values are reported. In the legend of table 1, “NS.” is mentioned, but “NS.” does not appear in the table.
Answer: we added the p values to table 1.
Reviewer 2 Report
The authors of this manuscript present the results of HEPBURN protocol, was designed as a multi-center randomized controlled trial to investigate the efficacy and safety of nebulized heparin versus placebo in patients with inhalation trauma.
Some issues need to be addressed to clarify this manuscript.
Major issues
Even the authors mention that the protocol was published (Trials. 2014 Mar 25;15:91. doi: 10.1186/1745-6215-15-91), I think would improve the reading and understanding to add from the HEPBURN protocol:
- Table 1. Clinical scoring and grading of inhalation trauma
- In Figure 1, you describe the Consort diagram with 116 expected patients randomization. I think would be good to mention this in the manuscript.
- Detailed inclusion and exclusion criteria
- Concomitant medication (allowed and not allowed), I think is good to add this information, as the routine concomitant use of N-acetylcysteine or mucolytics were excluded
Minor issues
- Figure 1 is not mentioned page 4
- Page 4. Table 1. Bronchoscopic, you give numbers in both columns, I don’t understand what it means
- Page 4, line 127. NS: not significant. I don’t see this result in table 1.
- Page 5, line 159. Control group, I think would be better use the term placebo, to be consistent in all manuscript.
- Page 5, lines 171, 176. Page 6, line 184. SAEs/SAE report. Does it mean Serious Adverse Event?, I don’t see this definition before
Authors responses
Comments and Suggestions for Authors
The authors of this manuscript present the results of HEPBURN protocol, which was designed as a multi-center randomized controlled trial to investigate the efficacy and safety of nebulized heparin versus placebo in patients with inhalation trauma.
Some issues need to be addressed to clarify this manuscript.
Major issues
Even the authors mention that the protocol was published (Trials. 2014 Mar 25;15:91.
doi: 10.1186/1745-6215-15-91), I think would improve the reading and understanding to add from the HEPBURN protocol:
Comment 1: Table 1. Clinical scoring and grading of inhalation trauma
Answer: Thank you for this comment. According to your suggestion we added additional information regarding the used inhalational injury severity scores (clinical and bronchoscopic) in the legend of table 1. We also refer to the Trials publication were the above-mentioned scoring systems are described in further detail.
‘Table 1:
a Clinical scoring of inhalation injury: Score consisting of 7 clinical factors considered to be suggestive for inhalation injury. One point each, diagnosis of smoke inhalation is fulfilled with a score > 2.
In brief:1. Trapped during a fire in an enclosed space, 2. Carbonaceous sputum, 3. Altered level of consciousness after the injury, 3. Mild respiratory distress, 4. Serious respiratory distress, 7. Hoarseness or loss of voice.
b Grading of inhalation injury by bronchoscopic criteria (0 = no injury; 1 = mild injury; 2 = moderate injury; 3 = severe injury; 4 = massive injury).
c Scoring based on chest X-ray findings, PaO2/FiO2, PEEP level and respiratory compliance. A score < 2.5 indicates mild-to-moderate lung injury, ≥ 2.5 indicates severe lung injury.’ (page 5, line 150-158)
‘… data on injury severity (including clinical and bronchoscopic inhalational injury severity scores (8)).’ (page 3 line 110-111)
Comment 2: In Figure 1, you describe the Consort diagram with 116 expected patients randomization. I think would be good to mention this in the manuscript.
Answer: according to this and other reviewers suggestion we added a paragraph with information regarding our sample size calculation.
‘Sample size calculation
Sample size calculation was based on a previous trial in which nebulized heparin was associated with a reduction of 4.6 days of mechanical ventilation in critically ill patients expected to require at least 48 hours of ventilation (10). We conservatively estimated a lower improvement in burn patients with inhalation trauma compared to nonburn critically ill patients and considered a reduction of three days of invasive ventilation as clinically significant. To observe an improvement of three ventilator-free days at day 28 with P < 0.05 at 80% power we required 58 patients per treatment arm (8).’ (page 3 line 116-122)
Comment 3: Detailed inclusion and exclusion criteria
Answer: we added detailed information regarding our inclusion and exclusion criteria in our methods section.
‘…Patients were excluded if … or were not expected to survive for more than 72 hours. Other exclusion criteria included: expected duration of mechanical ventilation of less than 24 hours, pregnancy or breast feeding, history of severe chronic obstructive pulmonary disease (e.g. with non-invasive ventilation or oxygen therapy at home; or need for frequent systemic corticosteroid therapy for exacerbations), pulmonary bleeding in the past 3 months, history of a significant bleeding disorder or a known allergy to heparin.’ (page 2, line 74-80)
Comment 4: Concomitant medication (allowed and not allowed), I think is good to add this information, as the routine concomitant use of N-acetylcysteine or mucolytics were excluded
Answer: Thank you for this suggestion, we added this information to our manuscript.
‘Routine use of mucolytics was not allowed and attending physicians were advised to use mucolytics only when viscous mucus was considered problematic. ‘ (page3, line 92-93)
Minor issues
Comment 1: Figure 1 is not mentioned page 4
Answer: we added the capture and legend of figure 1 on page 4.
Figure 1. Consort diagram
Abbreviations. TBSA: total body surface area. N: number
Comment 2: Page 4. Table 1. Bronchoscopic, you give numbers in both columns, I don’t understand what it means
Answer: All patients included in the study had inhalational injury confirmed by bronchoscopy. The severity of the inhalation injury was assessed using clinical and bronchoscopic grading scores. We added additional information regarding the scores used in the legend of table 1.
a Clinical scoring of inhalation injury: Score consisting of 7 clinical factors considered to be suggestive for inhalation injury. One point each, diagnosis of smoke inhalation is fulfilled with a score > 2.
In brief:1. Trapped during a fire in an enclosed space, 2. Carbonaceous sputum, 3. Altered level of consciousness after the injury, 3. Mild respiratory distress, 4. Serious respiratory distress, 7. Hoarseness or loss of voice.
b Grading of inhalation injury by bronchoscopic criteria (0 = no injury; 1 = mild injury; 2 = moderate injury; 3 = severe injury; 4 = massive injury).
c Scoring based on chest X-ray findings, PaO2/FiO2, PEEP level and respiratory compliance. A score < 2.5 indicates mild-to-moderate lung injury, ≥ 2.5 indicates severe lung injury.’ (page 5, line 150-158)
Comment 3: Page 4, line 127. NS: not significant. I don’t see this result in table 1.
Answer: we added the p-values to table 1, explaining that there are no significant differences.
Comment 4: Page 5, line 159. Control group, I think would be better use the term placebo, to be consistent in all manuscript.
Answer: thanks for this suggestion, we now use the term ‘placebo’ consistently throughout the manuscript.
Comment 5: Page 5, lines 171, 176. Page 6, line 184. SAEs/SAE report. Does it mean Serious Adverse Event? I don’t see this definition before
Answer: Indeed, it means ‘Serious adverse event’. We added the definition of this abbreviation to our manuscript.
‘… serious adverse events (SAEs) …’ (page 6, line 202).
Round 2
Reviewer 1 Report
Dear Editor,
as the comments from round 1 were answered, I have no further comments for the revised version.
Yours sincerely.
Reviewer 2 Report
The authors have put a considerable amount of effort in revising this manuscript, claryfying the information requested.